# Computational Method for Wavefront Sensing Based on Transport-of-Intensity Equation

**Iliya Gritsenko** [1,2]**, Michael Kovalev** [1,2]**, George Krasin** [1,2]**, Matvey Konoplyov** [1] **and Nikita Stsepuro** [1,*]

1. Laser and Optoelectronic Systems Department, Bauman Moscow State Technical University, 2nd Baumanskaya St. 5/1, 105005 Moscow, Russia; gritsenkoiv@student.bmstu.ru (I.G.); m.s.kovalev@bmstu.ru (M.K.); krasingk@lebedev.ru (G.K.); konoplevmo@student.bmstu.ru (M.K.)
2. Lebedev Physical Institute, Russian Academy of Sciences, Leninskiy Prospekt, 53, 119991 Moscow, Russia
* Correspondence: stsepuro@bmstu.ru

**Abstract:** Recently the transport-of-intensity equation as a phase imaging method turned out as an effective microscopy method that does not require the use of high-resolution optical systems and a priori information about the object. In this paper we propose a mathematical model that adapts the transport-of-intensity equation for the purpose of wavefront sensing of the given light wave. The analysis of the influence of the longitudinal displacement z and the step between intensity distributions measurements on the error in determining the wavefront radius of curvature of a spherical wave is carried out. The proposed method is compared with the traditional Shack–Hartmann method and the method based on computer-generated Fourier holograms. Numerical simulation showed that the proposed method allows measurement of the wavefront radius of curvature with radius of 40 mm and with accuracy of ~200 μm.

**Keywords:** wavefront; optical wavefield; phase distortions; wavefront sensor; computer modelling; transport-of-intensity equation





## 1. Introduction

To date, the majority of devices and systems for determining the surface relief, three-dimensional imaging, measuring and correcting aberrations, medical imaging and digital microscopy are based on methods for analyzing light fields by measuring intensity [1–5]. It is necessary to obtain all information about the light complex amplitude to characterize the light field, which is usually limited by the possibility of detecting only the intensity (amplitude) of the light beam. One of the most common ways to detect a phase argument of the complex amplitude is to convert the phase distribution to an amplitude distribution. The first to demonstrate such a possibility was L. Foucault, back in 1857, in his method known as the "Foucault knife-edge test" [6].

Wavefront sensors use the principles of geometric optics to transform the phase distribution into the intensity distribution [7–9]. Today the most commonly used wavefront sensor is a Shack–Hartmann sensor, which combines microlens array with a pixel detector. When the detector is placed in the focal plane of the microlens array, the position of the point focused by each microlens can be matched to a certain region of the wavefront. This method allows one to create a compact and simple scheme; therefore, Shack–Hartmann sensors are used in many fields of science [10–14]. However, processing the information recorded by the camera requires mathematical calculations to reconstruct the phase, which severely limits the performance and makes them applicable for certain tasks like atmospheric turbulence measurements only in pair with the state-of-art computers. It is also important that the parameters of the microlens array limit the spatial resolution, dynamic range and sensitivity of the sensor. Despite the fact that modern science has proposed several solutions to the abovementioned problems [15], technological processes still impose significant restrictions

on the size and radius of curvature (RoC) of the manufactured microlenses, which in turn limits the spatial resolution of commercial Shack–Hartmann sensors [16].

The use of holographic wavefront sensors is relevant in tasks requiring high performance [17–19]. In such sensors, the transformation of the phase distribution into the intensity distribution is carried out by filtering the light beams with holograms. Usually holographic wavefront sensors are used in the adaptive optical systems that require operating speed of more than a few kHz. Holographic sensors provide information about the wavefront in the form of a few tens of numbers—the amplitudes of aberration modes (for example, Zernike modes) or modes of adaptive mirrors. Thus, the signal from them can directly control the shape of the adaptive mirrors without additional processing.

In works [20,21], a method for measuring the wavefront aberrations of laser radiation based on computer-generated Fourier holograms was developed and experimentally verified. In this method, the process of measuring the wavefront is carried out via local optimization of the correlation function. In the future, devices implementing the described method can compete with interferometers and Shack–Hartmann sensors in problems where accurate measurements are required in a wide range of phase fluctuations, as well as at apertures larger than 10 mm.

All of the abovementioned methods use mathematical optimization algorithms to approximate the wavefront model. The optimization arguments are the weight coefficients of orthogonal basis functions for expanding wavefront in series, such as the Zernike, Chebyshev polynomials, Karhunen–Loève functions, etc. With an increase in requirements for accuracy, the number of orthogonal functions used to represent the wavefront also increases, which greatly reduces the processing speed. Increasing the data processing speed by finding an effective method for solving the optimization problem, using high-speed light modulators [22], as well as implementing algorithms for the synthesis of holographic filters using multi-threaded calculations remain as open problems in this area.

There are also non-interferometric methods based on some assumptions about the object wave and use mathematical algorithms to estimate information about the wavefront. Examples of such methods include Fourier-ptychography, wavefront curvature sensing [23,24] and pyramid wavefront sensors [25]. Removing the reference beam allowed to develop more simple and robust devices. Moreover, state of art computers have expanded the capabilities of such systems, increasing the field of view and achieving ultra-high resolution. At the same time, modern ptychographic microscopy experiments [26] show that to obtain an image of an object in a field of view of $50 \times 50$ $\mu m^2$ with a resolution of 34 nm at a wavelength of 10 nm, it takes 43 min of work of a personal computer.

In this regard, it is of interest to develop a non-interference method based on detecting the beam intensity in several planes, and numerically solving the transport-of-intensity equation to reconstruct the values of the complex field amplitude in the required plane in real time. One of the advantages of such a method is a removal of auxiliary optical components, usually used in other method for acquiring data/intensity distributions, so that the intensity distributions acquired directly on a camera. Like that one can avoid optical aberrations, which could be introduced by the auxiliary components. This article is focused on the wavefront sensing of the objectless light wave, namely, on measuring of the wavefront curvature of the spherical wave.

## 2. Reconstruction of the Phase of a Coherent Field via Transport-of-Intensity Equation

Modern methods of phase reconstruction, which are the development of ideas formulated in works 40 years ago, allow finding of a complex field on the surface of an object from the measured intensity distributions on the detector. In this case, the diffraction integral used in practice describes the field at the detector through the field distribution on the object surface. As a rule, the calculation of the diffraction integral is reduced to the calculation of direct and inverse Fourier transforms.

Let us show the role of the Fourier transform using the example of the transport-of-intensity equation, which describes the propagation of coherent radiation from an object to a detector, and is an exact solution of the wave equation.

It is known that a monochromatic coherent electromagnetic wave propagating in free space, in the absence of other sources, satisfies the Helmholtz equation

$$\left(\nabla^2 + k^2\right) U(r_\perp, z) = 0, \tag{1}$$

where $\nabla = \{\partial/\partial x; \partial/\partial y; \partial/\partial z\}$ is the Nabla operator for the three-dimensional space, $r_\perp = (x, y)$ vector radius in plane $(x, y)$, orthogonal to the direction $z$, $k$ is the wavenumber, $U(r_\perp, z)$ is the complex amplitude of the field of the light wave, which is expressed as

$$U(r_\perp, z) = \sqrt{I(r_\perp, z)} \exp\{j\phi(r_\perp, z)\}, \tag{2}$$

where $I(r_\perp, z)$ is the intensity of the wave, and $\phi(r_\perp, z)$ is its phase.

Using the complex field amplitude (2), we can obtain Equation (1) in the paraxial approximation

$$\nabla_\perp^2 U(r_\perp, z) + j2k \frac{\partial U(r_\perp, z)}{\partial z} = 0, \tag{3}$$

where $\nabla_\perp = \{\partial/\partial x; \partial/\partial y\}$ is the Nabla operator for the two-dimensional space.

Separating the real part of Equation (3), we obtain

$$\nabla_\perp \cdot [I(r_\perp, z)\nabla_\perp \phi(r_\perp, z)] = -k \frac{\partial I(r_\perp, z)}{\partial z}. \tag{4}$$

Equation (4) is called the transport-of- intensity equation (TIE). It connects the intensity $I$ and its longitudinal derivative $\partial I/\partial z$ with the phase $\phi$ of the light wave. Note that the TIE was originally derived from the wave equation [27], and later derived from Poynting's theorem [28] and other fundamental relations [29].

Teague [27] proposed one of the approaches to solving the resulting equation. Its essence lies in the application of the Helmholtz expansion theorem [30], which allows representing the vector field $I(r_\perp, z)\nabla_\perp \phi(r_\perp, z)$ in the form

$$I(r_\perp, z)\nabla_\perp \phi(r_\perp, z) = \nabla \psi(r_\perp, z) + [\nabla \times A(r_\perp, z)], \tag{5}$$

where $\psi(r_\perp, z)$ is the scalar potential, $A(r_\perp, z)$ is the vector potential. Neglecting the second term $[\nabla \times A(r_\perp, z)]$, expression (4) can be simplified into two standard Poisson equations.

$$\nabla_\perp^2 \psi(r_\perp, z) = -k \frac{\partial I(r_\perp, z)}{\partial z}. \tag{6}$$

$$\nabla_\perp \cdot \left[ I(r_\perp, z)^{-1} \nabla_\perp \phi(r_\perp, z) \right] = \nabla_\perp^2 \phi(r_\perp, z), \tag{7}$$

We can present Equation (6) with respect to the phase $\phi(r_\perp, z)$ [28,31] as

$$\phi(r_\perp, z) = -k\nabla_\perp^{-2} \left\{ \nabla_\perp \cdot \left[ \frac{1}{I(r_\perp, z)} \nabla_\perp \nabla_\perp^{-2} \frac{\partial I(r_\perp, z)}{\partial z} \right] \right\}. \tag{8}$$

One of the most popular methods for solving Equation (7) is the method based on the expression of differential operators in terms of the Fourier transform [31]. This solution is simple and effective. Let us consider the main idea of the method in more detail. Using the properties of the Fourier transform, one can express the sum of the partial derivatives of the $m$-th and $n$-th powers [32].

$$\frac{\partial^m f(x, y)}{\partial x^m} + \frac{\partial^n f(x, y)}{\partial x^n} = \mathcal{F}^{-1}\left\{ \left[ (jk_x)^m + (jk_y)^n \right] \cdot \mathcal{F}\{f(x, y)\} \right\}, \tag{9}$$

where $\mathcal{F}$ и $\mathcal{F}^{-1}$ are operators of direct and inverse Fourier transforms, respectively; $k_{x,y} = 2\pi\nu_{x,y}$ are the frequency coefficients, $\nu_{x,y}$ are the frequency grids.

Then the differential operators in Equation (8) can be written as

$$\nabla_{\perp}\{f(x,y)\} = \mathcal{F}^{-1}\left\{\left[jk_x + jk_y\right] \cdot \mathcal{F}\{f(x,y)\}\right\}, \tag{10}$$

$$\nabla_{\perp}^{-2}\{f(x,y)\} = \mathcal{F}^{-1}\left\{\left[\frac{1}{(jk_x)^2 + (jk_y)^2}\right] \cdot \mathcal{F}\{f(x,y)\}\right\}, \tag{11}$$

It should be noted that, for the application of relations (10) and (11), in view of the periodic nature of the Fourier transform, it is necessary to fulfill the periodic boundary conditions, under which the value of the function at the boundaries must be cyclically repeated [33].

## 3. Proposed Method

This section describes the mathematical model for determining the RoC of a spherical wave with a Gaussian intensity (Figure 1). We used the spherical wave as an object, which can be characterized by the RoC, while the gaussian intensity distribution is the most common distribution when it comes to coherent beams, such as laser beams. The novelty of the proposed approach is in the combination of two methods: TIE to reconstruct the phase of the complex field amplitude from a set of intensities and a geometric method (to be explained below) to calculate the RoC from a known complex field amplitude.

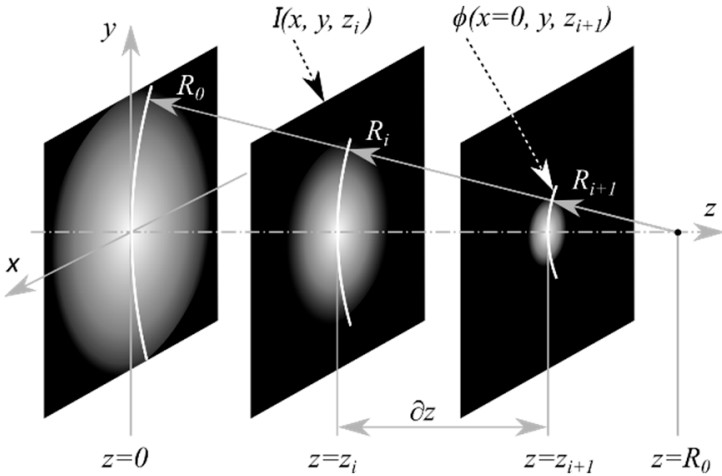

**Figure 1.** Propagation of a spherical wave in space.

Figure 2 presents a detailed representation of the modeling process, which consists of the following steps:

1.  Obtaining a set of complex amplitudes $U(r_{\perp}, z_i), \ldots, U(r_{\perp}, z_{i+1})$ by propagating the original field through space using the angular spectrum method [34], or acquiring a set of intensities $I(r_{\perp}, z_i)$ from measurements in a physical experiment (dashed block);
2.  Calculation of the theoretical RoC of the wavefront $R_t(z)$ using the geometric method;
3.  Calculation of phase components $\phi_{uw,TIE}(I, \partial I/\partial z_i)$ using only intensities $I(r_{\perp}, z_i)$ by solving the TIE (i.e., Equation (8));
4.  Calculation of RoC of the wavefront $R_{TIE}(z)$ using the initial intensities (step 1) and phases obtained in step 3 by the geometric method;
5.  Comparative analysis of $R_t(z)$ and $R_{TIE}(z)$.

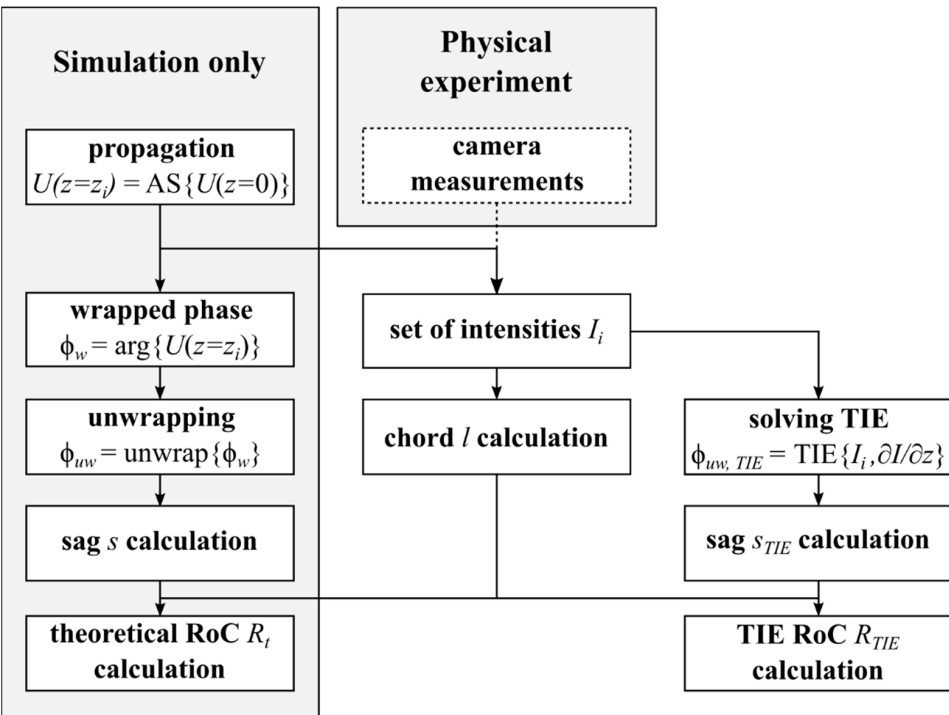

**Figure 2.** Flowchart with the steps of the modeling process. AS—angular spectrum method; TIE—transport-of-intensity equation; RoC—radius of curvature.

Next, the algorithm is explained more thoroughly. Note that to solve TIE (Equation (8)), it is necessary to know the longitudinal derivative with respect to the intensity $\partial I/\partial z$, which can be approximated by the finite difference method (Equation (12)), if at least two intensity distributions and the distance $\partial z$ between them are known (Figure 1).

$$\frac{\partial I(z)}{\partial z} = \frac{I(z_{i+1}) - I(z_i)}{\partial z}. \tag{12}$$

Therefore, the first step of the proposed algorithm (Figure 2) is to obtain a set of complex amplitudes $U(r_\perp, z_i), \ldots, U(r_\perp, z_{i+1})$ by propagating the initial field $U(r_\perp, z = 0)$ through space using angular spectrum method [34]. The propagation of the initial field is done only for simulation purposes: To attain intensities used later for TIE and for comparison in the end. For a physical experiment, acquiring a set of intensities in different planes will suffice.

At the second step, the theoretical radius of curvature $R_t$ is calculated using the geometric method. Let's consider it in more detail. It is known [35] that the radius of the circle $R$ is related to the sag $s$ and the chord $l$ by the following relation

$$R = \frac{s}{2} + \frac{l^2}{8s}. \tag{13}$$

The chord $l$ in our case is the width of the Gaussian function in level $I_{\max}(x)/e^2$ (Figure 3a). The search for the sag $s$ assumes that the complex field amplitude is zero where $I < I_{\max}/e^2$. Therefore, it is necessary to form an aperture with a diameter equal to the chord length $l$ and superimpose it on the wrapped phase $\phi_w$ (Figure 3b). The sag $s$ is found as the phase maximum after its unwrapping operation [36]. Thus, using Equation (13), one can determine the RoC of the wavefront.

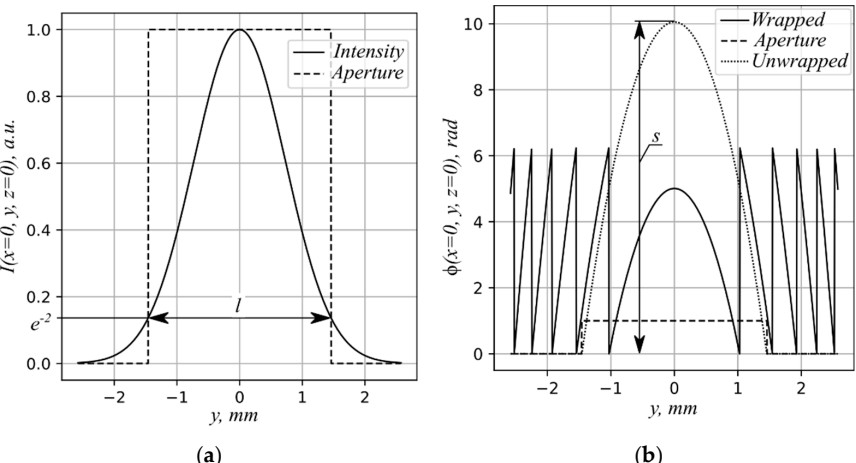

**Figure 3.** (**a**) Determination of the chord length l from the intensity of the modeled wave by level $I_{\max}/e^2$ for the synthesis of the aperture. (**b**) Overlay an aperture on a wrapped phase to obtain an unwrapped phase.

Note that the calculation of the theoretical RoC $R_t$ was done only for simulation purposes: To compare with the proposed method and to figure out its limitations and applicability. In a physical experiment there will be no calculation of the theoretical RoC. In this case other methods will be used to evaluate the performance of the proposed method (e.g., interferometry, Shack–Hartmann sensors and others).

Finally, in the third step of the algorithm, we find the derivative using Equation (12), solve TIE (Equation (8)) and get the already unwrapped phase $\phi_{uw,TIE}$ (one of the advantages of TIE). Next, we calculate the chord $l$ and the value of the sag $s_{TIE}$ and we calculate $R_{TIE}$ using Equation (13).

At the last step, a comparison of $R_t$ and $R_{TIE}$ is carried out, the results of which will be discussed in the next section.

## 4. The Limits of Applicability of Methods for Measuring the Curvature of the Wavefront

The proposed method is compared with the traditional method based on the Shack–Hartmann sensor and the holographic method using numerical simulation. The simulation was carried out in the Python programming language (http://www.python.org, accessed on 5 May 2021), which in recent years has become a powerful tool for fundamental and applied research in astronomy [37] and other related fields [38]. However, such comparison could be implemented in any other programming language (C++, MATLAB, etc.). For comparison, we took parameters of commercially available devices, such as Shack–Hartmann wavefront sensor and spatial light modulator with Full HD resolution.

### 4.1. Shack–Hartmann Sensor

Wavefront sensors are devices that measure the deviation of an optical wavefront from a plane or sphere. Let us calculate the minimum and maximum RoC that such a sensor can detect using the scheme shown in Figure 4.

The minimum radius $R_{\min}$ can be calculated from the condition of the maximum angle $\theta_{\max}$, by which one of the elements of the microlens array can deflect the incident beam, provided that the spot focused by the microlens does not fall on the area which is matched with another microlens. This angle is determined by the Equation (14):

$$\theta_{\max} = \frac{d}{2f}, \qquad (14)$$

where $d$ is the microlens diameter, $f$ is its equivalent focal length. For example, for a Thorlabs WFS 300-14AR sensor with $d$ = 300 μm and $f$ = 14.2 mm, the $\theta_{\max}$ value is defined as 10.56 mrad.

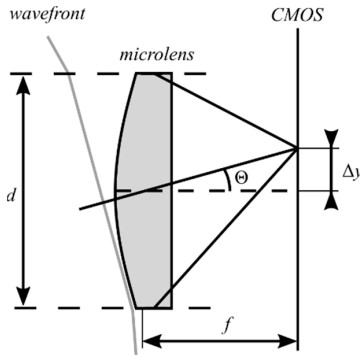

**Figure 4.** Position of the light spot in the microlens subaperture for a locally tilted wavefront.

The maximum radius $R_{max}$ can be calculated from the condition of the minimum detectable angle $\theta_{min}$, it is determined by the Equation (15)

$$\theta_{min} = \frac{\Delta y_{min}}{f}, \tag{15}$$

where $\Delta y_{min}$ is the minimum displacement of the light spot in the plane of the CCD that can be detected [39]. For CCD, it can be calculated as $\Delta y_{min} = a/20$, where $a = 4.65$ μm is the pixel size of the WFS 300-14AR sensor. Then we get that $\theta_{min} = 16.37$ μrad.

Based on the maximum and minimum angles, it is easy to calculate the minimum and maximum RoC that the sensor can register. The maximum radius $R_{max}$ can be calculated as follows

$$R_{max} = \frac{r_{det}}{\theta_{min}}, \tag{16}$$

where $r_{det}$ is the radius of the aperture in which the wavefront curvature is measured. The maximum radius of the aperture that can be inscribed in the WFS 300-14AR sensor aperture is 4.76 mm, then $R_{max} = 360$ m.

The minimum radius $R_{min}$ is calculated from the condition $\theta \leq \theta_{max}$, where $\theta$ is the angle by which the wavefront is deflected by the microlens located at the edge of the array. For the considered WFS 300-14AR sensor $R_{min} = 340.6$ mm. At the same time, proceeding from the accuracy of wavefront measurements ($\lambda/50$ for the selected WFS), we can determine the accuracy of the RoC measurement. The change of the sag $s$ (Equation (13)) of the local wavefront caused by increasing/decreasing of defocus value by $\lambda/50$ will result in changing of the RoC by 517 μm, while working around $R_{min}$.

Shack–Hartmann wavefront sensors output characteristic is linear on its entire dynamic range and its dynamic range is determined mostly by the microlens array used in the development. However, it was shown that the dynamic range can be expanded using different techniques. Namely the defocus aberration—-which can be directly translated to RoC of wavefront—-was measured with a maximum of 11% error over a range eight times larger than the microlens-bound definition [40]. However, such an error in estimation of the defocus value will translate to even bigger error in determining of the wavefront RoC ($\Delta R = 5$ mm for the RoC $R = 50$ mm for the parameters of the wavefront sensor given above).

### 4.2. Holographic Method Based on a Spatial Light Modulator

There is a compact device for measuring the wavefront based on highly efficient computer-generated holograms (CGH) in addition to traditional wavefront sensors [41,42]. The dimensions and resolution of the modulator used to output the CGH in such devices would determine the maximum spatial frequency of the CGH. Based on this, it is possible to determine the dynamic range of the measured values of the RoC for the method based on the spatial light modulator. According to [43], the resolution of the object recorded on the hologram should be four times less than the resolution of the final hologram. However, the expression is valid for the hologram recording in a physical medium. In the case when

the object is a CGH, it is enough to display it without distortion on the modulator. This can be achieved if at least 4 pixels of the hologram fall on one period of the interference pattern (structure) [44].

The CGH itself consists of two elements: the blazed diffraction grating and the phase function imposed on it. When measuring the RoC, the Zernike polynomial corresponding for defocus ($Z_2^2$), acts as a phase structure. In the limiting case, when work is carried out in zero order, the blazed diffraction grating can be neglected. The minimum RoC that can be registered without distortion can be calculated using the phase structure. With a modulator resolution of 1920 × 1080 pixels $R_{min}$ = 170 mm. With a further decrease in the radius on the final CGH, there are areas with a period of $T \leq 3$ pixels. The use of such CGHs will lead to an increase in the measurement error of RoC.

In general, the holographic method output characteristic is linear for a certain interval, which is determined by the values of the aberration modes, used during hologram synthesis [45,46]. Increasing the dynamic range by using the higher value of the aberration modes will decrease the accuracy of the method. The accuracy for wavefront aberration modes measurement vary from $\lambda/10$ [47] to $\lambda/50$ [18,21], which for the given parameters of the modulator (aperture 6912 mm for the pixel size of 6.4 μm) translates to RoC measurement accuracy $\Delta R$ = 201 μm and $\Delta R$ = 41 μm, respectively.

### 4.3. Method Based on the Transport-of-Intensity Equation

This section presents the results of three numerical simulations carried out using the mathematical model described in Section 3. In all numerical simulations, the following parameters were used: radiation wavelength $\lambda$ = 650 nm; field size $w \times h$ = 1500 × 1500 pixels; sampling step 5.04 μm. The intensity and phase in the $z = 0$ plane were specified by the following relations

$$I(x,y,0) = \exp\left\{-\frac{x^2 + y^2}{2l_0^2}\right\}, \tag{17}$$

$$\phi(x,y,0) = \sqrt{x^2 + y^2 + R_0^2}, \tag{18}$$

where $l_0$ = 750 pixels, $R_0$ value depends on the simulation.

In the first numerical simulation, the linearity of the proposed method was investigated for RoC $R_0$ = 100 mm, the range of distances $z = 0, \partial z, 2\partial z, \ldots, R_0$, where $\partial z$ = 1 mm. The obtained dependence (Figure 5) shows that the nonlinearity increases in direct proportion to the distance from the point $z = R_0$, that is, the error in determining RoC is proportional to the value of the chord $l$ (the width of the Gaussian function).

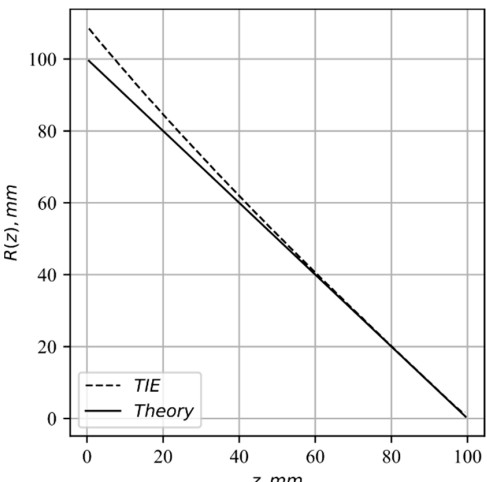

**Figure 5.** Dependence of the error in determining RoC of the wavefront on the $R_0$ = 100 mm; $z = 0$, $\partial z, 2\partial z, \ldots, R_0$; $\partial z$ = 1 mm.

During the second numerical simulation, the influence of the step value $\partial z$ was investigated for the following parameters: $R_0 = 100$ mm; $z = 80, 80 + \partial z, 80 + 2\partial z, \ldots , R_0$; $\partial z = 0.1, 1, 5, 10$ mm. Earlier [48] it was shown that the value of the step $\partial z$ should be large enough to cover the influence of noise and, at the same time, small enough so as not to violate the linearity of the derivative approximation (Equation (12)). In our simulation, there is no noise, so the step value is limited only from above, which is observed in the obtained dependence (Figure 6). More thorough study on the accuracy in the displacement of the between planes where intensity distributions were acquired was carried out in [49].

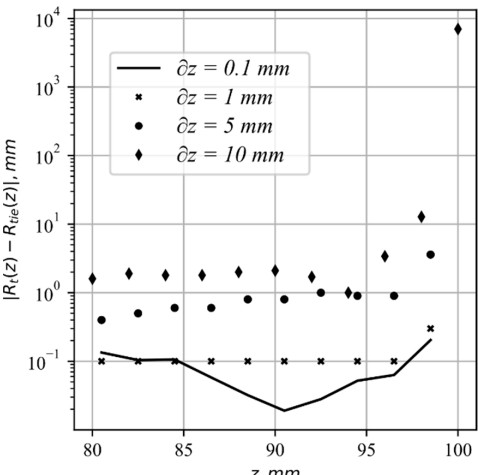

**Figure 6.** Dependence of the error in determining the RoC of the wavefront on the $\partial z = 0.1; 1; 5; 10$ mm.

Note that it is difficult to find the exact value of the chord $l$ near $R_0$ due to the effect of sampling. So, at $z = 97$ mm the chord $l = 5$ px, and the values in these 5 pixels differ from each other by $4 \cdot 10^2$ (for $I_{\max}(z = 0) = 1.0$). That is, to improve the accuracy in the region $z \cong R_0$, interpolation is necessary, which will allow obtaining a more accurate value of the chord $l$.

In the last numerical simulation, $R_{\min}$ was determined for the comparison with the methods presented above. The distance $(R_0 - z)$ was fixed: $(R_0 - z) = 15.5$ mm—that is, 15.5 mm in front of $R_0$; step $\partial z = 1$ mm; $R_0 = 20 \ldots 100$ mm. The resulting dependence (Figure 7) demonstrates the possibility of using this mathematical model at $R_{\min} \cong 40$ mm.

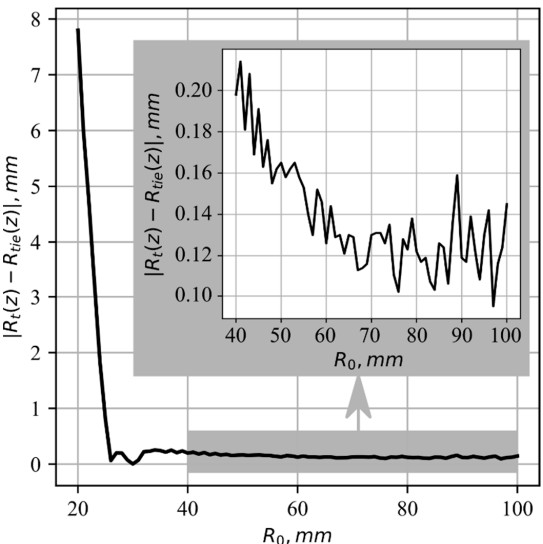

**Figure 7.** Dependence of the error in determining the RoC of the wavefront on the radius $R_0 = 20 \ldots$ 100 mm, $(R_0 - z) = 15.5$ mm, $\partial z = 1$ mm.

## 5. Results and Discussion

The main data from Section 4 was assembled in the Table 1. All of the errors were calculated based on the same radiation wavelength $\lambda$ = 650 nm. It can be seen, that while the proposed method does not achieve the highest accuracy of wavefront aberrations measurement, it still allows one to measure wavefront RoC with curvature of less than 60mm with adequate accuracy in comparison to other methods. So, we can say that the proposed method can expand the lowest range of wavefront RoC measurement, where other methods start showing high errors. In addition, all of this can be achieved with just a camera and software without the need of using focusing optics, which itself can introduce aberrations into the beam. Although it is worth noting that, for all the methods, the accuracy increases with an increase of RoC. That happens because the same change of the sag $s$ translates to smaller relative change in wavefront RoC. Figure 8 shows the graphical comparison of the dynamic range limits of the three methods.

**Table 1.** Comparison of wavefront radius of curvature measurement accuracy by different methods.

|  | Shack–Hartmann Wavefront Sensor | Holographic Wavefront Sensor | Proposed Method |
|---|---|---|---|
| Pixel Size, μm | 4.65 | 6.4 | 5.04 |
| Aperture Size | 4.76 | 6.91 | 7.56 |
| $R_{\min}$, mm | 50 | 170 | 40 |
| Defocus Measurement Accuracy | $10\lambda$ | $\lambda/50$ | $\lambda/1.5$ |
| $\Delta R$, μm | 5100 | 63 | 200 |

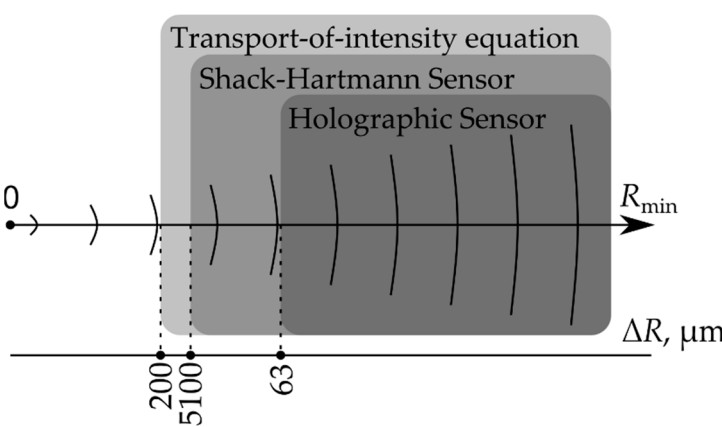

**Figure 8.** A visual comparison in the accuracy of determining the minimum RoC $R_{\min}$.

Certainly, there are devices with higher resolution (up to 4 and 8 K), that will allow to decrease the $R_{\min}$, while increasing the accuracy. Likewise, the same changes could be done to the proposed method. Thus, even using more common commercially available devices one can achieve better results while implementing the proposed method. In this work we showed the principle of the proposed method for the case of the wavefront RoC measurement with symmetric wavefront profile (defocus aberration mode). However, for the TIE it does not matter if the wavefront profile is symmetric or asymmetric as long as the wavefront function satisfies the boundary conditions. Naturally the asymmetry will impact the RoC calculation process, but the algorithm can be adapted by introducing the polynomials into the calculation process [50].

## 6. Conclusions

In this study we proposed a method for wavefront sensing of the light radiation based on the transport-of-intensity equation. The proposed method was studied using

numerical simulation and compared to the common devices, such as Shack–Hartmann wavefront sensors and Holographic wavefront sensors. The comparison was done with the radius of curvature of the wavefront of spherical wave. The resulting error in measuring the wavefront radius of curvature $R_0$ = 40 mm at $\lambda$ = 650 nm was 0.20 mm. The proposed method enables to expand the dynamic range of the radius of curvature of the wavefront measurement.

The novelty of the proposed method lies in the fact that the transport-of-intensity equation was for the first time used to directly measure the wavefront of the light wave (i.e., wavefront sensing), unlike its previous uses in microscopy imaging. The estimates obtained in the article could be used in the development of optoelectronic systems designed to register, process and display information about optical radiation.

**Author Contributions:** Conceptualization, methodology, project administration and funding acquisition, M.K. (Michael Kovalev); software, I.G. and M.K. (Matvey Konoplyov); visualization and writing—original draft preparation, I.G.; resources, investigation, writing—review and editing, N.S. and G.K.; All authors have read and agreed to the published version of the manuscript.

**Funding:** This research was funded by Russian Science Foundation, grant number 20-79-00264.

**Data Availability Statement:** The data is available from the corresponding author upon request.

**Acknowledgments:** The authors thank Nina Verenikina for productive discussions.

**Conflicts of Interest:** The authors declare no conflict of interest.

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
