# Peer review of "Computational Method for Wavefront Sensing Based on Transport-of-Intensity Equation"

_photonics, doi:10.3390/photonics8060177_

Round 1

Reviewer 1 Report

Dear Authors,

Thank you for taking my remarks into account. In my opinion the paper has become better, but some more points should be clarified before its' acceptance for publication. Please, find the comments attached.

Reviewer 2 Report

The authors have improved the manuscript, and now it is a little bit more clear what they are doing and why they think it is worth investigating. What I found confusing in the first draft was that it seemed that they do experiments and have somehow obtained the complex field which they then use in the algorithm to retrieve the complex field with some error.

From the improved manuscript it is more clear now that there is nothing experimental being done, and the complex field they propagate is for demonstration purposes only (if I have understood correctly). Therefore, I would suggest that this point is made more clear and, for example, in Fig. 2 the flowchart could be modified such that the first step (propagation of the complex field) is somehow completely separated from the proposed algorithm so that potential readers do not confuse it to be an integral part of the algorithm. Additionally, it would be helpful to point out in the text that the complex field is propagated only to attain intensities used for TIE and the phase for comparison at the end.

It would also seem that the novelty aspect of the manuscript has been addressed and the results support the conclusion that the proposed method is an improvement. Therefore I recommend acceptance after minor revision.

Reviewer 3 Report

The authors greatly improved the quality of the paper. In the overall, they answered to most of the points indicated in the first revision. However, the main problem subsists: its minor level of novelty. Although the authors now demonstrate some degree of novelty, they compare only with “non-TIE-based methods” (traditional Shack-Hartmann and computer-generated Fourier hologram methods), while there are several TIE-based methods (e.g. see https://ascimaging.springeropen.com/articles/10.1186/s40679-016-0017-y, or https://doi.org/10.1016/j.optlaseng.2020.106187, or https://www.bnl.gov/isd/documents/89005.pdf) that could have been mentioned in the state of the art and compared to the proposed method, particularly regarding its differences and position of the proposed methods face to them.

The field of applicability seems also limited, although they prove to achieve a good balance between the method’s simplicity and performance. In lines 311-312 the authors mention that for the TIE it does not matter if there’s symmetry or not, but what about the impact of asymmetry in the proposed calculation of the theoretical radius of curvature using the geometric method? And if the beam does not have a Gaussian intensity distribution?

So, although there is a significative increase in the quality of the proposed paper, and some originality can be found, I still believe it is not enough for it to be accepted to Photonics.

Reviewer 4 Report

All my objections have been answered and the manuscript is much improved and ready for publication

Author Response

We sincerely thank the Reviewer for the time and positive assessment of the work.

Kind regards, Authors

Round 2

Reviewer 3 Report

The authors presented valid arguments regarding the previous review. Thus, I believe it can be accepted for publication, but it’s quality would improve if the authors can include the core of their answers also in the text to be published. In particular, the answer to point 2 (response 2) regarding the asymmetry and gaussian beam nature of the study would be important for the readers clearly understand the scope and potential of the work.

Author Response

This manuscript is a resubmission of an earlier submission. The following is a list of the peer review reports and author responses from that submission.

Round 1

Reviewer 1 Report

Dear Authors,

I believe that the manuscript can be published in Photonics, but some more work should be done. Please, find my comments in the attachement.

Reviewer 2 Report

The manuscript titled "Object-oriented modeling of the phase problem of optics: algorithm, capabilities and limitations" by Gritsenko et al., seems attempt to establish an algorithm for computational imaging (?) via the transport of intensity equation. However, there are several serious problems with the manuscript, and it cannot be published.

First, It is not clear what the authors wish to do. Is it imaging? Phase retrieval? Wavefront sensing? All of these are of course related, but they are not synonymous.

Second, the authors make several contradicting statements. For example, starting from line 92, they remark how the manuscript is devoted to going beyond paraxial methods, and yet, the equations they employ rely on the paraxial approximation (equation (3) onwards).

Third, the proposed method seems to require the authors to first propagate the complex field with the angular spectrum approach. If you can use the angular spectrum approach, then you do not need to computationally propagate the intensity! The whole outline of their algorithm is in fact unclear.

Fourth, the authors state that they have done some experiments, yet they do not offer almost any experimental details. What was the employed source? Which quantities did they measure? And what do they retrieve?

All in all, the present manuscript does not qualify for publication. Therefore I must recommend rejection.

Reviewer 3 Report

Please see the comments in the attached pdf file

Reviewer 4 Report

The authors address the topic of phase imaging and reporting their work in implementing the phase reconstruction using the transport-of-intensity equation (TIE). They implemented a model in a specific programming language (Python) and simulate the methodology applied to a point source.

Although the paper has no major problems regarding the language used or the quality and meaning of the images, there are a few points that must be mentioned:

  1. It is not clear what is the novelty of the work being presented. Although a rather generic title, at the end, what it is presented is just a computational implementation of an algorithm to solve the TIE, with the simulation of a (simple) example of a point source and error analysis. The latter is interesting, but rather limited regarding what is presented and the current state of the art on the subject.
  2. In the last paragraph of section 1, the authors point out “one of the main advantages” of the method as being its ability to “overcome the problems associated with aberrations (…) and approach the diffraction limit of resolution (…)”, but their results do not address this (very) important point.
  3. In the introduction (lines 85 to 87), the programming language (Python) is mentioned as a “powerful tool”. However, there is no further development on its use, and eventual relevance, in the scope of the work being presented. In fact, after the introduction, it is not mentioned elsewhere. At the end, does it matter if the algorithm is implemented in Python, C++ or even MatLab? As it concerns, the programming language must be mentioned but as a “material” of the methodology and not in the introduction as a relevant tool.
  4. The explanation of the flowchart presented in fig. 2 (lines 153 to 172) can be improved. Points 1 and 2 are too much separated from the remaining points due to the detailed explanation in lines 157 to 167. The authors can have this explanation after resuming the steps.
  5. In the section regarding “the limits of applicability of methods for measuring the curvature of the wavefront”, besides its number should be 4 and not 3, three methods are compared: the Shack-Hartmann sensor, the holographic method based on a special light modulator (SLM), and the TIE-based method. This is an interesting analysis, but it could be highly improved. A broader analysis should be made instead of applying individual examples to each method. That is, common operational parameters should be defined, and the limits analysed in that scope. Also, the performances of the different methods should have been compared directly, for example, using a table.

On the overall, and concluding, besides having several important points that should be addressed by the authors, the main issue is the novelty of the work being presented. For all the mentioned reasons, and because I believe that just a major revision will not overcome it, I recommend its rejection.

Reviewer 5 Report

As in the attached Word document
